# Meta-16S rRNA Gene Phylogenetic Reconstruction Reveals the Astonishing Diversity of Cosmopolitan Myxobacteria

**DOI:** 10.3390/microorganisms7110551

**Published:** 2019-11-11

**Authors:** Yang Liu, Qing Yao, Honghui Zhu

**Affiliations:** 1State Key Laboratory of Applied Microbiology Southern China, Guangdong Provincial Key Laboratory of Microbial Culture Collection and Application, Guangdong Open Laboratory of Applied Microbiology, Guangdong Microbial Culture Collection Center (GDMCC), Guangdong Institute of Microbiology, Guangdong Academy of Sciences, Guangzhou 510070, China; liuhaolin3@163.com; 2College of Horticulture, South China Agricultural University, Guangdong Province Key Laboratory of Microbial Signals and Disease Control, Guangdong Engineering Research Center for Grass Science, Guangdong Engineering Center for Litchi, Guangzhou 510642, China; yaoqscau@scau.edu.cn

**Keywords:** myxobacteria, phylogenetic diversity, geographical distribution, environments

## Abstract

Numerous ecological studies on myxobacteria have been conducted, but their true diversity remains largely unknown. To bridge this gap, we implemented a comprehensive survey of diversity and distribution of myxobacteria by using 4997 publicly available 16S rRNA gene sequences (≥1200 bp) collected from several hundred sites across multiple countries and regions. In this study, the meta-16S rRNA gene phylogenetic reconstruction clearly revealed that these sequences could be classified into 998 species, 445 genera, 58 families, and 20 suborders, the great majority of which belonged to new taxa. Most cultured myxobacteria were strongly inclined to locate on the shallow branches of the phylogenetic tree; on the contrary, the majority of uncultured myxobacteria located on the deep branches. The geographical analysis of sequences based on their environmental categories clearly demonstrated that myxobacteria show a nearly cosmopolitan distribution, despite the presence of some habitat-specific taxa, especially at the genus and species levels. Among the abundant suborders, Suborder_4, Suborder_15, and Suborder_17 were more widely distributed in marine environments, while the remaining suborders preferred to reside in terrestrial ecosystems. In conclusion, this study profiles a clear framework of diversity and distribution of cosmopolitan myxobacteria and sheds light on the isolation of uncultured myxobacteria.

## 1. Introduction

Myxobacteria are a large group of organisms, which taxonomically belong to the order *Myxococcales* within the class *Deltaproteobacteria* [1]. They are extensively dispersed in natural environments, preferentially in places that are rich in microbes and organic matter. Myxobacteria are one of the most fascinating and unique bacteria by virtue of their sophisticated social lifestyle and distinctive characteristics [2,3], including cooperative swarming [4], group predation [5], multicellular fruiting body formation [6], and sporulation. Under favorable environmental conditions, vegetative cells of myxobacteria glide on a solid surface in swarms. According to nutritional behavior and specialization in the degradation of biomacromolecules, myxobacteria are divided into two distinctive groups: micro-predators capable of lysing living microbial cells by excreting multiple lyases, and cellulose-decomposers [7]. More notably, except for a few strains such as *Myxococcus xanthus*, most strains could form species-specific and colorful fruiting bodies by oriented cell movement after nutrient depletion. Fruiting bodies contain a substantial number of desiccation-resistant myxospores, which can survive in hostile environments and are able to germinate under appropriate conditions even after decades of resting [8]. A further intriguing feature of myxobacteria is their outstanding capability to produce structurally diverse bioactive secondary metabolites [9]. Over the past decades, more than 100 new carbon skeleton metabolites and over 600 derivatives have been identified from thousands of myxobacterial strains. These metabolites include polyketides, non-ribosomal peptides, and hybrids thereof, and exhibit remarkable antifungal, antibacterial, cytotoxic, antiviral, immunosuppressive, antimalarial, and antioxidative activities with different modes of action [10]. Therefore, myxobacteria have proven to be a promising source for new bioactive metabolites. In addition, some myxobacteria have been determined to lyse pathogenic bacteria and fungi [11], produce diverse carotenoids [12], degrade 2-chlorophenol [13], and reduce uranium(VI) [14], clearly demonstrating their wide potential applications in biomedicine, agriculture, industry, and environmental protection.

As a phylogenetically coherent and distinctive group, members of the order *Myxococcales* have been divided into three suborders, mainly based on morphological properties of cells and sequences analysis of the 16S rRNA gene; for instance, *Cystobacterineae*, *Sorangiineae*, and *Nannocystineae* [15]. At the time of starting this study, the monophyletic order *Myxococcales* comprised 11 families, 29 genera, and 66 species enlisted in the List of Prokaryotic Names with Standing in Nomenclature database (LPSN, http://www.bacterio.net/, last accessed November 2018). In addition to a minority of type strains, the majority of non-type strains have been isolated from diverse ordinary and extreme habitats, and preserved in multiple culture collections or research teams, for example, more than 9000 myxobacteria were collected by Reichenbach and Höfle [16]. Compared with cultured myxobacteria, there are more uncultured types, which were easily checked and identified via multiple culture-independent approaches such as clone libraries [17] and high throughput sequencing of 16S rRNA gene amplicons [18]. These culture-independent approaches bypassed isolation and cultivation procedures and provided information on the in situ abundance and diversity of myxobacteria. These research findings clearly showed the existence of many novel phylotypes of phenotypically undescribed myxobacteria along with known phylotypes. As a result, it may be inferred that myxobacteria in nature should show much higher diversity than what has been revealed so far. However, the true diversity of myxobacteria has remained largely unknown until now, due to some reasons such as heavy dependence on morphological properties, less attention on the uncultured strains, the inadequate numbers of 16S rRNA gene sequences in the phylogenetic diversity of myxobacteria.

Myxobacteria are widespread in terrestrial ecosystems [19]. Historically and until recently, soils, decomposing plant materials such as tree bark, rotting wood, and leaves of trees, and dung of herbivores have always been considered as typical environments of these organisms [20]. Myxobacteria have also been isolated from limnetic habitats [18]. According to the explanation for this phenomenon by Reichenbach [19], they are not actually indigenous limnetic organisms, and originally come from terrestrial surroundings before being transported into water bodies. Surprisingly, myxobacteria have also been found in some extreme conditions, such as acidic wetlands [21], saline-alkaline soils [22], arid deserts [23], and even hot springs [24]. More unexpectedly, a few myxobacteria have sporadically been isolated from marine habitats [25], and are clearly different from known terrestrial myxobacteria, particularly in their requirement for sodium chloride roughly equivalent to the salinity of seawater [26]. Since this discovery, marine myxobacteria have attracted much attention worldwide. The study by Jiang et al. showed that myxobacteria-related sequences from four deep-sea sediments and a hydrothermal vent were diverse but phylogenetically similar at different locations and depths [17]. They were separated from terrestrial myxobacteria at high levels of classification matching a phylogenetic distance of approximately 10%, likely resulting from geographic separation and environment selection [17]. Soon after, Brinkhoff et al. demonstrated that an obligate marine myxobacteria cluster (MMC) constituted up to 13% of total bacterial 16S rRNA genes in the surface sediments of the North Sea, and was also detected in most sediment samples from other sea areas [27]. These results described above suggest that unique habitats may harbor specialized taxa of myxobacteria owing probably to the hypothetical local adaptation. Inspired from these studies, we hypothesized that myxobacteria could be more taxonomically diverse than what is acknowledged presently, due to their enormously diverse, distinct niches at a global scale. However, the detailed relationships between taxa and different environments and the distribution patterns of myxobacteria have not yet been thoroughly investigated.

With the development of next-generation sequencing technologies and the increasing number of studies on myxobacteria, thousands of 16S rRNA gene sequences of myxobacteria have been rapidly accumulated in public databases. These available sequences thus provide an ideal opportunity for attempting to answer the above-mentioned questions. In the current study, we first collected and integrated 16S rRNA gene sequences from the multiple databases. Phylogenetic analysis based on myxobacteria-related sequences was then performed to infer their taxonomic diversity. Using this extensive survey, we addressed the following questions: (i) How many taxa of cultured and uncultured myxobacteria at different taxonomic levels are there so far? (ii) What is the environmental distribution pattern of myxobacteria inhabiting diverse environments worldwide? (iii) What are the ecological preferences of individual myxobacterial lineages? Our discoveries will provide new insights into the diversity and ecological distribution of myxobacteria inhabiting diverse environments worldwide, and will help in isolation of the uncultured myxobacteria and screening of bioactive metabolites.

## 2. Materials and Methods

### 2.1. Dataset Construction of 16S rRNA Gene Sequences of Myxobacteria

A considerable number of 16S rRNA gene sequences of myxobacteria with various lengths have been accumulated in multiple public databases including the National Center for Biotechnology Information (NCBI) database, SILVA database [28], RDP database [29], and Integrated Microbial Genomes (IMG) database [30]. We retrieved 16S rRNA gene sequences from the four databases above. For the NCBI database, a preliminary search using the keyword of “Myxococcales [ORGANISM] AND 16S [TITLE]” in the nucleotide database was performed. To obtain enough genetic information and to embrace more available sequences, near-full-length (≥1200 bp) sequences (hereafter defined as Dataset 1, the same below) were picked out from candidate items. The 16S rRNA gene sequences from the complete and draft genomes of myxobacteria were collected as the Dataset 2 by using blast genome alignment with default algorithm parameters unless otherwise specified. In this process, 16S rRNA gene of type strain *Myxococcus xanthus* DSM 16526^T^ (accession number: DQ768116.1) was used as a query sequence, since the strain has been used as a model organism for studies on myxobacteria and is near the middle of the myxobacterial phylogenetic tree based on 16S rRNA gene sequences. According to the fact that identities among all validly published type strains of the order *Myxococcales* are above 80% and to obtain as many myxobacterial sequences as possible, the slightly lower identity of 78% and 90% coverage were chosen. Both multiple-copies and single-copy sequences from the respective genome were used in the following analysis. Thousands of sequences (Dataset 3) were searched and downloaded according to their taxonomic context in the SILVA database (release 132). The target sequences (Dataset 4) were obtained from the RDP database under these options including the type and non-type strains, uncultured and cultured isolates, with the sequence length at or above 1200 bp, good quality, and nomenclatural sequences. The sequences (Dataset 5) were acquired from the IMG database by using the BLAST databases including 16S rRNA public isolates and public assembled metagenomes with an E-value of 1e-5 on 29 October 2018. All sequences from the five databases were integrated as a raw dataset.

### 2.2. Sequences Validation and Environmental Categories of Myxobacteria

The sequence validation was carried out according to the following rigorous criteria: (1) The longest high-quality sequence was selected as the representative of the same strain with different original numbers; (2) the ambiguous base ratio of each sequence was less than 0.2%, the equivalent of less than three ambiguous bases in 1500 bases; (3) the sequences which do not belong to the order *Myxococcales* were excluded based on the screening of the RDP classifier with more than 80% confidence; (4) chimeric sequences were identified by the free package USEARCH61 [31] and then removed. Finally, a total of 4997 high-quality 16S rRNA gene sequences including those of all type strains were analyzed with the GenBank accession numbers and IMG gene IDs indicated in Appendix A. The definition of cultured and uncultured myxobacteria was taken from user-provided metadata or literature retrieval. The isolation sources of all sequences were retrieved from the original annotations, published studies, or documents of culture collections, and then manually assigned to multiple defined environmental categories referring to the previously described schema [32,33]. To simplify categorizations, sequences from marine animals, marine plants, marine biofilms, and others were classified as sequences from marine organisms, whereas those from terrestrial animals, terrestrial plants, terrestrial biofilms, and others as sequences from terrestrial organisms. Notably, sequences from activated sludge and human skin, although in terrestrial ecosystems, were divided into two different categories, in order to highlight their uniqueness and distinctiveness. Altogether, 10 environmental categories were designated as Marine_organism (hereafter abbreviated as M_organism, the same below), Marine_sediment (M_sediment), Seawater, Soil, Terrestrial_sediment (T_sediment), Activated_sludge, Terrestrial_organism (T_organism), Freshwater, Human, and The_unknown (the unknown isolation sources), as shown in Appendix A. The isolation geographical coordinates and countries of sequences were also determined according to user-provided metadata or literature retrieval as much as possible. Rarefaction curves of all samples were calculated by using the software Mothur version 1.38 [34].

### 2.3. Phylogenetic Analysis of 16S rRNA Gene Sequences of Myxobacteria

To analyze the phylogeny of the 16S rRNA gene, all sequences were aligned by using the software MAFFT version 7.037b with default settings (https://mafft.cbrc.jp/alignment/software/) [35]. The approximately maximum-likelihood (ML) phylogenetic tree of 16S rRNA gene sequences was drawn by using the default model of GTR + CAT and 1000 bootstrap replicates within the software FastTree version 2.1.10 [36]. In order to make the display more intuitive and concise, bootstrap values were only indicated by solid circles filled by light blue (rgba(200, 200, 255, 0.8)) with different sizes ranging from 0.5 to 3 px in the middle (50%) of branches of the phylogenetic tree in Appendix A. To compare with previous studies, the sequence of *Desulfovibrio desulfuricans* ATCC 27774 (accession number: M34113), a sulfate-reducing bacterium also in the class *Deltaproteobacteria* [37], was chosen as an outgroup to root the phylogenetic tree. The visualization, annotation, and management of phylogenetic trees were performed by using the web-based tool of Interactive Tree Of Life (iTOL) [38].

The taxonomic thresholds are the vital cornerstone of the phylogenetic analysis of myxobacteria. In this study, to realize maximum congruence for myxobacterial phylogeny, we drew lessons from the general taxonomic thresholds from the studies by Yarza et al. [39,40], the thresholds to a large extent matching the family level from the studies by Garcia et al. [41] and Jiang et al. [17], and the mean, median, and range of identities among type strains of the order *Myxococcales*. After integrating these values, the taxonomic hierarchy of all sequences was determined based on the conservative criteria below: The most widely used identity of 97.0% or lower for two 16S rRNA gene sequences serves as strong evidence for distinct species, 94.5% or lower for distinct genera, 89.0% or lower for distinct families, and 85.0% or lower for distinct suborders. For the sake of comparison, the taxonomic affiliation of all type strains was re-identified based on the above criteria. The pairwise identities of aligned sequences were calculated by using the software DNAMAN version 8 (Lynnon Biosoft, San Ramon, CA, USA, https://www.lynnon.com/) with the distance method of Kimura. Combined with phylogenetic analysis and the above taxonomic thresholds, the classification of each sequence was carried out with artificial modification, if necessary. In addition, in order to ensure the repeatability and reproducibility of all figures, the Hex values and rgb/rgba values of colors used to mark different elements including suborders, environmental categories, the culturability, lineages, and so forth, are also provided in Appendix A.

## 3. Results

### 3.1. The Isolation Sources and Locations of 16S rRNA Gene Sequences of Myxobacteria

In this study, we have collected a large number of sequences from diverse environments. The 16S rRNA gene sequences used in this study were from PCR amplicons, clones, genomes, and metagenome assemblies, respectively. For isolation sources, sequences from the soil, marine organisms, marine sediments, activated sludge, and terrestrial organisms accounted for 59.42%, 9.11%, 6.92%, 6.80%, and 5.08% of the sequences, respectively, while the remaining sequences from others sources only contributed 12.67% (Appendix A). The isolation locations of sequences with the exact longitudes and latitudes are marked by red dots with black rings in Figure 1, most of which are situated in East Asia, South Asia, Europe, and North America. The sequences were obtained from more than 80 countries and areas (Figure 1 and Appendix A). More than 80% of the sequences were from 10 countries, including China, Denmark, the USA, Mexico, Panama, Japan, India, Germany, Spain, and France. A small number of sequences came from the oceans, such as the Pacific Ocean and the Atlantic Ocean. The results showed that the sampling is indeed very broad but still biased towards East and South Asia, Europe, and North America.

### 3.2. The Unexpected Diversity of Myxobacteria

A highly resolved phylogenetic tree of 16S rRNA gene sequences clearly presented an unexpected taxonomic diversity of myxobacteria (Figure 2, Appendix A, and Appendix A). Specifically, myxobacteria were divided into 20 suborders, 58 families, 445 genera, and 998 species according to phylogenetic positions and taxonomic criteria mentioned above (Table 1). At the suborder level, 20 suborders were comprised of the three previously described suborders (*Cystobacterineae*, *Sorangiineae*, and *Nannocystaceae*) [41] and 17 newly discovered suborders (Suborder_1 to Suborder_15). In this study, inspired by definitions of abundant and rare species [42], the abundant and rare suborders were defined based on the proportion of 16S rRNA gene sequences, with ≥0.5% classified as an abundant suborder and <0.5% as a rare suborder. Under this criterion, abundant suborders included *Sorangiineae*, *Cystobacterineae*, *Nannocystaceae*, Suborder_16, Suborder_15, Suborder_17, and Suborder_4, accounting for 29.08%, 26.58%, 20.89%, 10.67%, 4.72%, 4.14%, and 2.88% of all sequences, respectively (Table 1). Rare suborders included the remaining 13 taxa, namely, Suborder_1 to Suborder_3 and Suborder_5 to Suborder_14, accounting for 1.04% of all sequences altogether. Thus, abundant suborders seemed more common than rare ones.

A highly resolved phylogenetic tree of 16S rRNA gene sequences clearly presented an unexpected taxonomic diversity of myxobacteria (Figure 2, Appendix A, and Appendix A). Specifically, myxobacteria were divided into 20 suborders, 58 families, 445 genera, and 998 species according to phylogenetic positions and taxonomic criteria mentioned above (Table 1). At the suborder level, 20 suborders were comprised of the three previously described suborders (*Cystobacterineae*, *Sorangiineae*, and *Nannocystaceae*) [41] and 17 newly discovered suborders (Suborder_1 to Suborder_15). In this study, inspired by definitions of abundant and rare species [42], the abundant and rare suborders were defined based on the proportion of 16S rRNA gene sequences, with ≥0.5% classified as an abundant suborder and <0.5% as a rare suborder. Under this criterion, abundant suborders included *Sorangiineae*, *Cystobacterineae*, *Nannocystaceae*, Suborder_16, Suborder_15, Suborder_17, and Suborder_4, accounting for 29.08%, 26.58%, 20.89%, 10.67%, 4.72%, 4.14%, and 2.88% of all sequences, respectively (Table 1). Rare suborders included the remaining 13 taxa, namely, Suborder_1 to Suborder_3 and Suborder_5 to Suborder_14, accounting for 1.04% of all sequences altogether. Thus, abundant suborders seemed more common than rare ones.

At the family level, the dominant families constituting more than 10% of all sequences were Cystobacterineae_Family_6 (21.83%), Sorangiineae_Family_5 (16.75%), and Sorangiineae_Family_4 (12.13%); the subdominant families constituting more than 5% were Nannocystaceae_Family_11 (5.74%) and Suborder_16_Family_4 (5.66%); the remaining 53 families contained relatively few myxobacteria-related sequences (Appendix A). At the genus level, Cystobacterineae_Family_6_Genus_27 (15.47%) was the most dominant genus; Sorangiineae_Family_4_Genus_21 (6.80%) and Sorangiineae_Family_5_Genus_19 (4.62%) were the second and third most dominant genera; each remaining genus constituted less than 3% of the total sequences (Appendix A). At the species level, Cystobacterineae_Family_6_Genus_27_Species_5 (12.55%), Sorangiineae_Family_4_Genus_21_Species_1 (6.80%), and Sorangiineae_Family_5_Genus_19_Species_3 (4.54%) were dominant species relative to other non-dominant species (each less than below 3%, Appendix A). These results indicated that myxobacteria present an incredible diversity compared to that expected from previous studies and were biased to some taxa listed in Appendix A at the three taxonomic levels from family to species.

A comparison between the numbers of taxa identified in this study and those previously identified at each taxonomic level was conducted. According to the newly conservative criteria above, the validly described type strains of the order *Myxococcales* were reclassified into four suborders, seven families, 13 genera, and 24 species (Appendix A). Therefore, we concluded that 16 new suborders, 51 new families, 432 new genera, and 974 new species were determined in this study. The occurrence frequencies (%) of the new families, genera, and species within the three known suborders were compared based on the taxonomic determination of myxobacteria in this study. As shown in Appendix A, abundant new taxa at taxonomic levels from family to species were found among the three suborders. The high occurrence frequencies of new taxa were shown, especially those of new genera and species above 90%. Rarefaction is one of the most widely used methods to assess species richness from the results of sampling. Here, we constructed two types of rarefaction curves for all 16S rRNA gene sequences based on abundant suborders and environments. The significant differences of sequence numbers, whether at seven abundant suborders (the number from 144 to 1453) or at ten environments (the number from 72 to 2969), result in the unsuitability for data normalization and further comparative analysis with each other. Therefore, we only focused on the trend of each rarefaction curve. Both suborder-based and environment-based rarefaction curves were far from the plateau at 97% identity (Figure 3), clearly indicating that more species could be found in either each suborder or each environmental category as more sequences are checked. The patterns were also supported by the coverage values of “samples” (Appendix A). Therefore, the results explicitly demonstrated that the currently defined taxa of the order *Myxococcales* are just the tip of the iceberg.

### 3.3. The Expanded Phylogeny of Myxobacteria

After the determination of the taxonomic diversity of myxobacteria, the evolutionary relationships were also inferred from the phylogenetic tree of 16S rRNA gene sequences. As shown in Appendix A, myxobacteria consisted of three well-supported lineages, designated as lineage A, lineage B, and lineage C. More specifically, the minor lineage A congruent with Suborder_1 was phylogenetically distant from the other two lineages, and located at the base of the tree, theoretically suggesting that lineage A was early deep branching. Intriguingly, the five strains of lineage A resided in distinctive habitats located in different geographic regions, namely the deep-sea sediment of the Pacific Ocean, soil in Brazil, periphyton in China, rice paddy field soil in Japan, and cave water in the USA. The lineages B and C were more closely related to each other than to lineage A, and thus were considered as sister lineages. The medium lineage B holding 169 myxobacteria-related sequences was on the equivalent of three suborders, from Suborder_2 to Suborder_4. The largest lineage C consisted of six abundant suborders and 10 rare ones. In lineage C, Suborder_5 occupied the basal-most position; the next five suborders from Suborder_6 to Suborder_14, which contained 22 strains, were later and sub-basal taxa; the remaining six suborders were phylogenetically young taxa based on their position and dominant taxa according to the high proportion of 96.1% of all sequences.

Within these isolates, there were 3890 uncultured myxobacteria, 1101 cultured ones, and six with few indications as to whether they have been cultured or not. With an exception of type strain DSM 53668^T^ (CP011125), all cultured strains were affiliated with the three known suborders, which was indicative of their easy-to-culture characteristics, and vice versa for uncultured strains. For isolation environments from soil to human, as shown in Appendix A, the percentages of the cultured strains decreased gradually, and conversely, those of the uncultured strains increased successively, implying that the expected probabilities of finding new genotypes or strains from the latter environments are much greater than those from the former. More intriguingly, we observed that cultured myxobacteria including the well-described type strains, were overall inclined to locate in the shallow branches of the phylogenetic tree; on the contrary, those of the uncultured myxobacteria were inclined towards the deep branches (Appendix A). Therefore, the culturability of myxobacteria basically matched branching patterns in the 16S rRNA-based phylogenetic tree.

### 3.4. Global Biogeographic Distribution of Myxobacteria

To determine the geographical distribution patterns of myxobacteria, the suborder-level sequences were summarized and compared according to isolation environments. As shown in Figure 1 and Appendix A, myxobacteria were widely distributed in both terrestrial and marine environments, thereby suggesting a cosmopolitan distribution in these habitats. Considering their small number of sequences, the distribution of the rare suborders was not further analyzed in depth. The myxobacteria within each abundant suborder contained all environmental categories. Nevertheless, as shown in Figure 4a, they were subjected to uneven distributions across environments to some degree. More specifically, *Sorangiineae*, *Cystobacterineae*, and *Nannocystaceae* were mainly from terrestrial environments, especially from the soil. On the contrary, Suborder_4, Suborder_15, and Suborder_17 were dominant in marine environments, including marine organism, marine sediments, and seawater. Moreover, the distribution of sequences at the family level was also unequal in multiple environments, as illustrated in Figure 4b. The results shed light on the possibility of isolating myxobacteria of specific taxa from specific habitats. For example, we are more likely to isolate strains of Family_5 and Family_6 in Suborder_17 from the hypersaline mat (Appendix A), which belonged to the M_organism. Strikingly, the two previously neglected environments including activated sludge and human skin also contained a certain number of myxobacteria within each abundant suborder. Furthermore, more than half of the myxobacteria from activated sludge belonged to the suborder *Nannocystaceae*. On the contrary, myxobacteria from human skin were scattered over the seven abundant suborders.

A comprehensive analysis of the relationships between myxobacterial taxa, including generalists (ubiquitous, cosmopolitan taxa) and specialists (environment-specific taxa) and different environments at the four taxonomic levels was performed. In this study, we defined a specificity criterion as having 80% or more of their observations (≥5) of single taxon belong to a single environmental category, otherwise, it was defined as a cosmopolitanism. As illustrated in Figure 4c, the specificity of myxobacteria showed an increasing trend from the suborder to species levels, whereas cosmopolitanism showed the reverse trend. It is also worth noting that the specificity of taxa is closely related to environmental categories and taxa number. As the number of environmental categories or taxa increases, specificity may decrease or disappear, even at the species level. Certainly, according to the current criterion, habitat specificity of some taxa at the family, genus, and especially species levels, were evidently shown in this study (Figure 1, Appendix A), including the known MMC cluster matching the genus Suborder_16_Family_4_Genus_33 from marine sediments.

## 4. Discussion

Compared to many in-depth studies on the physiological basis of social behavior and on action mechanisms of secondary metabolites of myxobacteria over the last several decades, much less attention has been paid to their taxonomic diversity and biogeographic distribution across heterogeneous environments. In this case, by using the public 16S rRNA gene sequences of myxobacteria, we performed a comprehensive and systemic review to expand our understanding of the two aspects above. These findings demonstrate that myxobacteria from various environments have remarkable taxonomic diversity and a widespread geographic distribution, which is far beyond that expected based on previous studies.

From the discovery of myxobacteria to the present, an increasing number of bacteria have been isolated using different media and culturing conditions. In the course of the study, on the correlations between morphological characteristics and 16S rRNA gene phylogeny of myxobacteria, three lineages/suborders (*Cystobacterineae*, *Nannocystineae*, and *Sorangiineae*) were determined via branching pattern of the phylogenetic tree [15]. About 10 years later, the study by Garcia et al. not only reaffirmed the three suborders of myxobacteria, but also discovered nine new taxa, which were probably at the family level, based on the expanded phylogeny of 16S rRNA gene sequences of 101 myxobacteria [41]. With the rapid development of sequencing technology, a growing number of uncultured myxobacteria have been detected by using culture-independent methods, such as the high-throughput 16S rRNA amplicon sequencing and metagenomic sequencing. Among these ecological surveys, the great diversity of myxobacteria has been documented in varied ecological niches [17,18], and more meaningfully, some new habitat-specific clusters have been found, such as the MMC cluster, as mentioned above [27,43,44]. However, from the perspective of classification, the real phylogenetic diversity of cultured and uncultured myxobacteria is still unknown, and an accurate taxonomic affiliation for most new taxa has not yet been determined to date. Here, the present study demonstrates that the thousands of myxobacteria are highly diverse at the four taxonomic ranks from species to suborder, most members of which are defined as novel taxa based on 16S rRNA gene phylogenetic reconstruction. Moreover, we provided a relatively complete and stable evolutionary framework of myxobacteria for classification of currently described new taxa and subsequent cultured and uncultured strains, especially those distantly related to valid type strains. Just recently, the positive correlation between taxonomic distance and the production of distinct secondary metabolite families was substantiated through a systematic metabolite survey of about 2300 myxobacteria [16], which provided the important strategic implication that the use of bacteria belonging to new species, genera, families, and even suborders could substantially increase the discovery chance of new natural products, and further highlights the importance of the myxobacterial classification system within this scenario. Therefore, a more detailed taxonomic system of myxobacteria was established in the current study, which could fuel the discovery of interesting novel compounds in the foreseeable future.

Similar to actinobacteria and bacillus, myxobacteria have been considered for a long time as typically terrestrial organisms [20]. Now, along with the more phylogenetic analyses of myxobacteria from marine environments, it has been widely accepted that some distinctive marine myxobacteria are widespread in upper oxic sediments and bottom anoxic sediments from different locations and depths [17,27], which are phylogenetically distinct from soil and limnetic bacteria [45]. In addition to natural marine sediments [46], myxobacteria as the second abundant group also presented in oil-polluted subtidal sediments, with their relative abundance decreasing through depth [47]. Somewhat incomprehensibly, a few myxobacteria are only sporadically found in oxic seawater samples [27], which is consistent with our result of the presence of only about 1% seawater bacteria. On the contrary, Ganesh et al. described metabolically active myxobacteria as representing up to 3% of the total sequences in the larger size fraction from a marine oxygen minimum zone [48]. These results indicate that strict or facultative anaerobic myxobacteria, which have been of little concern for a long time, are also dominant in marine environments, just like their well-described aerobic relatives. More studies looking into how myxobacteria live in oxygen-free environments and what the metabolic differences between aerobes and anaerobes are need to be further elucidated.

Meanwhile, some myxobacteria reside in less focused environments, such as activated sludge [49,50] and microbial mats/biofilm [51]. In the study by Wang et al. [52], the order *Myxococcales* is one of the core orders of the microbial community, with a range from 0.86% to 9.5% of the classified sequences from activated sludge samples. However, the role of myxobacteria in these biological wastewater treatment systems has not been well studied. Unexpectedly, more than 3% of the myxobacteria in this study were from the skin of children with atopic dermatitis [53,54]. Therefore, we suggest to all myxobacteriologists that more attention should be paid to the genetic diversity and population structure of myxobacteria from human-related samples. Altogether, consistent with the report by Tamames et al. [32], two possible geographical patterns for myxobacteria have been determined in this study: Most myxobacteria at the high taxonomic ranks of family and suborder could share many environments; and some at the species and genus levels could be found specifically within each environment. The underlying adaptation mechanisms for pandemic and endemic distribution patterns of myxobacteria under different taxonomic hierarchy still need to be explored.

At present, the isolation, purification, and cultivation of myxobacteria are the biggest challenges, and are still essential for elucidating myxobacterial physiology in depth. Numerous strategies have been successfully used to increase the cultivation efficiency of myxobacteria as well as to isolate novel taxa; for example, isolation of salt-tolerant myxobacteria from marine conditions [25]. There is no doubt that the present study may provide preliminary clues for the culture of previously uncultured members of myxobacteria. On the one hand, after determining the biogeographical distribution of myxobacteria, we can select environmental samples purposefully rather than randomly and blindly to isolate target bacteria, despite the fact that isolation efforts may be in vain. For example, when the MMC is targeted, the chance of successful isolation from marine sediments is much greater than that from other habitats [27]. Based on this study and those by others, the previously neglected habitats with an astonishing diversity of myxobacteria appear to promising reservoirs for obtaining yet-to-be-cultured bacteria. On the other hand, our study clearly indicates that isolation efforts under anaerobic conditions may be a more effective alternative for obtaining myxobacteria [55]. The characterization and description of the species *Anaeromyxobacter dehalogenans* as the first facultative anaerobic myxobacteria has proven the feasibility of anaerobic cultivation approaches [13]. Additionally, based on the fact of the MMC being restricted to salinities ranging from six to 60 practical salinity units [27], salinity is another very important factor and thus should be given more attention in the process of isolating myxobacteria.

## 5. Conclusions

In this report, we contributed a far more comprehensive understanding of the expanded diversity and biogeographic distribution pattern of myxobacteria from diverse environments, which was based on the phylogenetic reconstruction of thousands of 16S rRNA gene sequences. The myxobacteria analyzed in this study are considerably diverse, with a wide distribution in terrestrial and marine environments, and harbor abundant new taxa at each taxonomic rank. This study structures a robust taxonomic framework of myxobacteria and provides overarching guidance for innovation of isolation techniques and the discovery of new secondary metabolites.

## Figures and Tables

**Figure 1 microorganisms-07-00551-f001:**
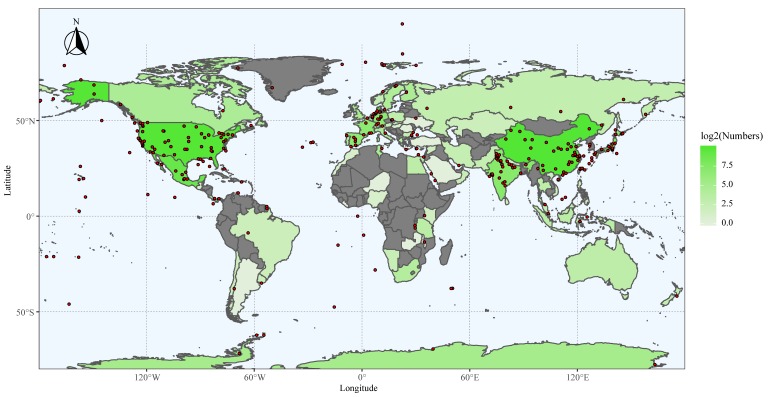
Map of the geographical distribution of the strains, showing the isolation locations marked by red dots with black rings, the countries, and areas. The countries and areas are marked by gradient colors from “#E2EFDA” to “#54E635” based on the logarithmic values of sequence numbers. Some countries in grey with a few sequences are not illustrated in the map and are shown in Appendix A. The graph was plotted by using the “ggplot2” package with the geom.sf() function (https://cran.r-project.org/web/packages/ggplot2/index.html) according to the longitude and latitude coordinates of isolated sites. The world map was generated by using the “rworldmap” package (https://cran.r-project.org/web/packages/rworldmap/index.html).

**Figure 2 microorganisms-07-00551-f002:**
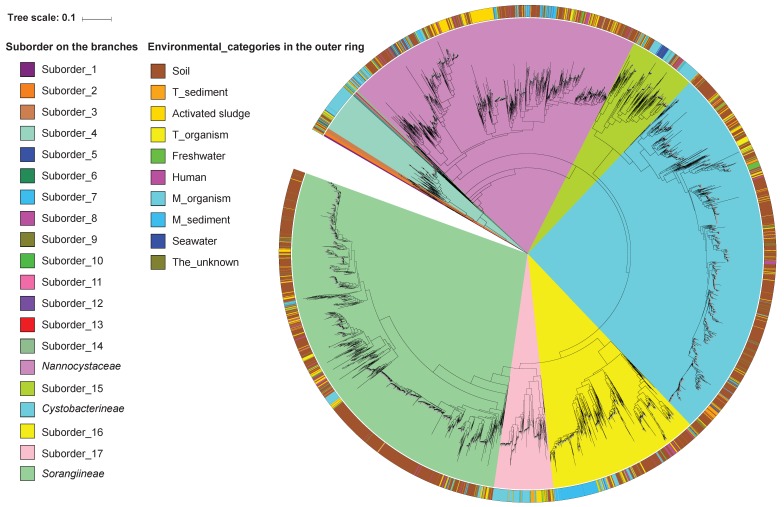
The phylogenetic tree inferred from 16S rRNA gene sequences showing the positions of bacteria within the order *Myxococcales*. The sequence of *Desulfovibrio desulfuricans* ATCC 27774 roots the tree. Scale bars: 0.1 substitutions per nucleotide position. Suborder-level clusters are indicated by 20 different colors (as shown in Appendix A) on the branches of the tree, and environmental categories of all sequences are indicated by the 10 different colors in the outer ring of the tree.

**Figure 3 microorganisms-07-00551-f003:**
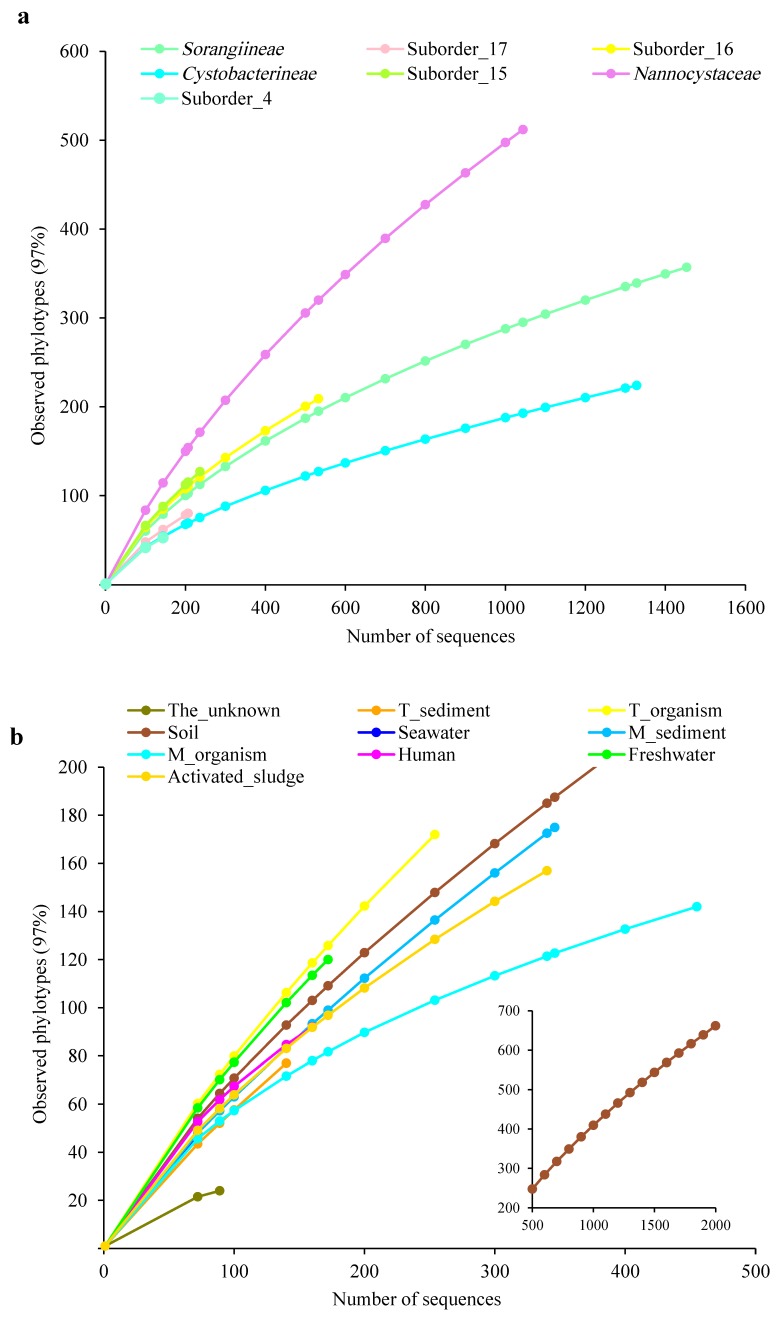
Rarefaction curves for all 16S rRNA gene sequences based on the seven abundant suborders (**a**) and 10 environmental categories (**b**). Operational taxonomic unit s was determined at 97% identity. The colors of points and curves in the figure are listed in Appendix A.

**Figure 4 microorganisms-07-00551-f004:**
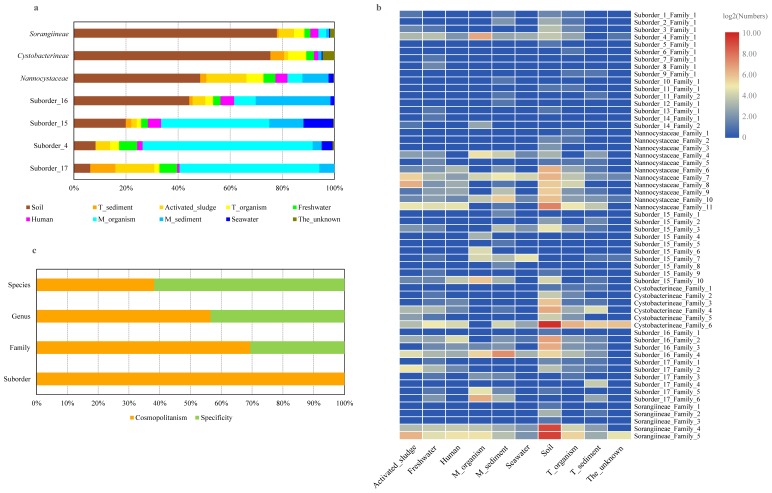
The distribution of the seven abundant suborders (**a**) and 58 families (**b**), and the proportion of specific and cosmopolitan taxa at the four taxonomic levels (**c**). The graph in right panel was visualized by using the “heatmap” function in the R software version 3.5.1 with default parameters.

**Table 1 microorganisms-07-00551-t001:** The detailed taxonomy of myxobacteria at the four taxonomic levels.

Abundant/Rare	Suborder	Family	Genus	Species	Numbers (%)
Rare	Suborder_1	1	5	5	5 (0.10)
Rare	Suborder_2	1	3	6	10 (0.20)
Rare	Suborder_3	1	3	8	15 (0.30)
Abundant	Suborder_4	1	28	44	144 (2.88)
Rare	Suborder_5	1	1	1	1 (0.02)
Rare	Suborder_6	1	1	1	1 (0.02)
Rare	Suborder_7	1	1	1	1 (0.02)
Rare	Suborder_8	1	1	1	2 (0.04)
Rare	Suborder_9	1	2	2	2 (0.04)
Rare	Suborder_10	1	1	1	1 (0.02)
Rare	Suborder_11	2	3	3	4 (0.08)
Rare	Suborder_12	1	1	1	1 (0.02)
Rare	Suborder_13	1	2	2	2 (0.04)
Rare	Suborder_14	2	3	3	7 (0.14)
Abundant	*Nannocystaceae*	11	157	309	1044 (20.89)
Abundant	Suborder_15	10	48	108	236 (4.72)
Abundant	*Cystobacterineae*	6	50	131	1328 (26.58)
Abundant	Suborder_16	4	54	116	533 (10.67)
Abundant	Suborder_17	6	32	60	207 (4.14)
Abundant	*Sorangiineae*	5	49	195	1453 (29.08)
Total	20	58	445	998	4997

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
