# Peer review of "Meta-16S rRNA Gene Phylogenetic Reconstruction Reveals the Astonishing Diversity of Cosmopolitan Myxobacteria"

_microorganisms, 2019, doi:10.3390/microorganisms7110551_

Round 1

Reviewer 1 Report

This study presents an updated phylogeny of the Myxobacteria based on 16s gene sequences. It explores the correlation between the taxonomic diversity and the geographical distribution of the myxobacteria, and proposes a criteria to estimate the degree of cosmopolitanism. I appreciated the neat effort made by authors to make their study reproducible. I believe it is an interesting piece of work that should be published after minor revisions.

Main comments:

- the main claim of increased described diversity compared to previous studies should be better explained as part of the introduction. We don't know clearly what is the difference between those previous studies and the present one. Even more importantly is remains unclear what does the "newly rigorous criteria above" (line 241) bring to the study compared to the (unexplained) previous criteria. I think it is very important to inform the reader in the introduction about the importance of this new phylogeny. Some material is already present in the discussion (l 336-363) but should be explained earlier in the manuscript.

- The working definition of diversity remains somewhat unclear, as well as the definition of a bacterial species (knowing all issues related to this concept in bacteria). The definitions the authors are using for this particular study should be clarified.

Additional comments:

- l 43: M. xanthus starts forming fruiting bodies before the food source is depleted.
- l 71: "based on the numerous research findings, we inferred that (...)". This is a very strong statement that has no solid bases. You should cite these "research findings" and explain clearly why it leads you to the hypothesis that most diversity is still unknown in Myxobacteria.
- l 96: Does this statement rely on an hypothetical local adaptation? I think this part should be clarified.
- l 105-108: I appreciate the clear statement of the questions. However, the second question is unclear to me. What is the environmental distribution pattern?
- l 135: the second criterion is unclear.
- l 144: what is the previously described schema?
- l 160: How did you choose the ML model?
- l 162: the blue circles for bootstrap values seem like a good idea but it is almost impossible to see the different sizes on the figure.
- l 181: The title of this section is not very informative.
- Fig 1: Are the countries in grey not sampled at all? It should be explained in the legend.
- The sampling is indeed very broad but still biased towards Asia, Europe and North America. This should be more directly stated.
- l 246: I don't understand "sufficently clear taxonomic determination of myxobacteria".
- l 294: How did you "summarize" the sequences?
- Fig 4: I could not find the information related to the heatmap construction. What software and what settings did you use?
- The information you get from the rarefaction curves should be better explained.
- The discussion contains interesting information but the structure could be improved to better highligh the new insights from this study compared to previous work.
- I could not see tables S1, S5 and S9 but that might be an issue from my side.
- The manuscript would greatly benefit from English language and style editing.

Reviewer 2 Report

The manuscript by Liu et al describes a computational analysis of 16S gene diversity associated with the Myxococcales. 16S sequences are compiled from various databases and used to generate an iTOL tree. The authors argue for the existance of numerous new taxa within the Myxococcales and argue that some taxa are environment-specific.

The manuscript has paragraphs which are very well written, but many more that are poorly expressed, with inappropriate and non-scientific use of terms and a general lack of technicality. The text must be edited for English-language usage. For example how are the following terms defined?: endowed (line35), some (line 186), huge, extensive (line 192), impressive (line248), sharp (line 250), overrepresented (line 239), dominant (line 239), perfectly comparable (line 290), summarized (line294)

Many of the arguments and approaches are circular or not adequately justified.

Identification of myxobacterial 16S sequences was based on BLAST results using a DSM16526 query and 90% coverage and 78% identity. Why was this query chosen, and why were those cutoffs selected?

Figure 1. Clearly myxobacteria have been isolated from all over the world, but the relative numbers of isolates reflect the number of researchers rather than any aspect of myxobacterial prevalence or diversity. It does nothing to ‘adequately demonstrate broad representativeness of sequences’.

I would argue that the taxonomic diversity represented in Figure 2 is entirely expected, with most isolates belonging to previously identified clades. Proposal of new taxa is entirely dependent on criteria of 97%, 94.5% and 85% identity. These need to be justified. The Yarza et al approach to classification is still contentious. Even their publications acknowledge that to give maximum congruence with existing taxa with standing in nomenclature, different cut-offs need to be applied for different taxa, especially, but not exclusively, with pathogens. Therefore these cut-offs need justification, and they should be tweaked to give maximum congruence with known taxonomy. Most new taxa proposed are deeply branching and have few members and no cultured representatives. Much more evidence than a few 16S sequences is required to propose them as novel sub-orders, especially in the myxobacteria, which have long been accepted as requiring a polyphasic taxonomic approach. As the rarefaction curves illustrate, more sequences means more taxa and with an exponential increase might never be expected to plateau. The apparent diversity is to be expected.

Line 217-218. This is indeed so obvious that it should be removed. Of course commoner taxa are more common than rarer taxa. There is no basis presented to describe taxa as ‘overrepresented’. What would their representation be expected to be. You wouldn’t expect a flat distribution between taxa and would therefore expect some to be dominant compared to others.

Section 3.3 The basis by which the tree is broken into 3 lineages is not presented. Many more lineages could have also been proposed equally arbitrarily. Lineage A is not a common ancestor of all myxobacteria. It merely represents an early deep branching within the taxa.

Lines 287. Taxa that are more heavily sampled have more sequences, and are more likely to have similar sequences in the set, and therefore will have shorter branch lengths. The difference in branch length is just an inevitable consequence of including both culturable and unculturable organisms.

Lines 383-384. It takes much more evidence than a (low) %presence of myxobacteria to propose myxobacteria cause dermititis. This goes beyond speculation.

Round 2

Reviewer 2 Report

Thanks to the authors for improving the manuscript.

The explanations provided in response to points 2 and 4 are fine, but that reasoning should have been included in the manuscript for the readership to see.

point 8 - I still think it goes too far to even suggest any potential link between myxobacteria and health.
